# Perceived Facilitators and Barriers to Nigerian Nurses’ Engagement in Health Promoting Behaviors: A Socio-Ecological Model Approach

**DOI:** 10.3390/ijerph17041314

**Published:** 2020-02-18

**Authors:** Chinenye Uchendu, Richard Windle, Holly Blake

**Affiliations:** 1School of Health Sciences, University of Nottingham, Nottingham NG7 2RD, UK; 2NIHR Nottingham Biomedical Research Centre, Nottingham NG7 2UH, UK

**Keywords:** nurses, health promoting behaviors, facilitators, barriers, socio-ecological model, Nigeria, diet, physical activity, alcohol consumption, smoking

## Abstract

Nurses make up the single largest healthcare professional group in the Nigerian healthcare system. As frontline healthcare providers, they promote healthy lifestyles to patients and families. However, the determinants of Nigerian nurses’ personal health promoting behaviors (HPBs) remain unknown. Utilizing the socio-ecological model (SEM) approach, this study aimed to explore the perceived facilitators and barriers to Nigerian nurses’ engagement in HPBs. HPBs were operationalized to comprise of healthy dietary behaviors, engagement in physical activity, low-risk alcohol consumption, and non-smoking behaviors. Our study was carried out in a large sub-urban tertiary health facility in Nigeria. Data collection was via face-to-face semi-structured interviews and participants were registered nurses (*n* = 18). Interview data were transcribed verbatim and analyzed thematically to produce nine themes that were mapped onto corresponding levels of influence on the SEM. Findings show that in Nigeria, nurses perceive there to be a lack of organizational and policy level initiatives and interventions to facilitate their engagement in HPBs. The determinants of Nigerian nurses’ HPBs span across all five levels of the SEM. Nurses perceived more barriers to healthy lifestyle behaviors than facilitators. Engagement in healthy behaviors was heavily influenced by: societal and organizational infrastructure and perceived value for public health; job-related factors such as occupational stress, high workload, lack of protected breaks, and shift-work; cultural and religious beliefs; financial issues; and health-related knowledge. Organizations should provide facilities and services to support healthy lifestyle choices in Nigeria nurses. Government policies should prioritize the promotion of health through the workplace setting, by advocating the development, implementation, regulation, and monitoring of healthy lifestyle policies.

## 1. Introduction

Nurses constitute the largest occupational group of frontline healthcare professionals in the health sector. Nurses provide 24-h care and experience prolonged direct contact and engagement with individuals and their families. For optimum health, wellness and role performance, it is essential for nurses to engage regularly in healthy lifestyles. Pender’s health promotion model defined health-promoting behaviors (HPBs) as health-related actions directed at increasing an individual’s level of wellness, self-actualization, and wellbeing [1,2]. In this context, HPBs include healthy eating, engagement in regular physical activity, abstinence from smoking, and low-risk/non-alcohol consumption. Regular engagement in HPBs constitutes a healthy lifestyle and significantly impacts on the health status of individuals. However, previously published research evidence shows that nurses’ HPBs include unhealthy dietary behaviors, physical inactivity, risky levels of alcohol consumption, and smoking [3,4,5,6,7,8,9,10,11,12,13]. This indicates that nurses may not be adequately transferring their health-related knowledge into practice in their personal lives. The International Council of Nurses (ICN) has also outlined a statement based on concerns about nurses’ HPBs (ICN) [14]:
“*If each of the world’s 13 million nurses made a personal commitment to eat healthily, exercise appropriately, and avoid the use of tobacco, this would improve their health and well-being and reduce the likelihood of their developing chronic diseases.*”(ICN 2010: p41)

The global concern for nurses’ HPBs also applies in Nigeria. Nigeria is currently the most populated country in Africa with over 195 million people [15], yet the nurse-population ratio is low at 1.5:1000 [16]. The health work force of Nigeria is an index for determining the quality of care that is available to the population and affects its global health ranking. Although Nigeria is one of the countries with highest number of human resources for health in Africa [17], out of 49 low income countries identified by world bank parameters as having markedly low numbers of nurses and doctors, Nigeria is sixth in position and falls well below the critical threshold of 23 nurses per 10,000 population [18,19]. According to the World Health Organization, the current trends of health workforce recruitment and retention in the African region will be significantly insufficient to tackle the health needs of the population by the year 2030 [20]. As a result, strategies to promote healthy lifestyles among Nigeria nurses may hold potential for improving their health and well-being, but firstly, the determinants of nurses’ HPBs need to be investigated. Nigerian nurses have been previously reported to experience job dissatisfaction, which was significantly associated with their working conditions and hospital environment [21]. There is a nursing shortage in Nigeria, which is a huge burden to the Nigerian healthcare system, population health, and nurses’ health and wellbeing. International and urban migration of nurses further contributes to acute nursing shortages and inequity in the distribution of nurses countrywide [22,23]. In recognition of the increasing demand for healthcare and nursing services as a result of an increase in the aging population and continuous increase in Nigeria’s population, Nigerian nurses’ HPBs are a primary concern for this study. Concerns around nurses’ lifestyle behaviors are a priority because the structures and processes available in Nigeria’s health sector expose nurses to working in an ‘unhealthy reality’ [24,25]. Unhealthy lifestyles due to poor working conditions may result in adverse health effects among Nigerian nurses, although little is actually known about the health and wellbeing of Nigerian nurses or the reasons for their lifestyle choices, since this has not previously been explored. Currently there is only anecdotal evidence of potentially problematic health profiles and occupational challenges to health and wellbeing. Furthermore, an unhealthy nursing workforce in Nigeria may worsen already existing nursing shortages and retention challenges experienced in this country, and limit the availability and quality of healthcare provision nationwide, which is already well below expected standard. Therefore, it is critical to understand the enablers and barriers to nurses’ personal practice of HPBs to support a healthy Nigerian public health workforce for the future; these enablers and barriers may align with issues raised in other settings or be specific to the Nigerian context. Nurses in Nigeria work at the primary, secondary, and tertiary levels of healthcare. In the Nigerian context, nurses mostly deliver primary and secondary healthcare services within the community. However, nurses at the tertiary health services regularly deliver preventive health interventions (such as lifestyle related health education) to patients as part of their roles. In addition, as a result of urbanization, dissatisfaction with primary health services and preference for care delivery by multidisciplinary teams of health professionals, more people utilize tertiary health facilities in Nigeria compared to primary health facilities [26]. Previous international research studies on facilitators and barriers to nurses’ HPBs found that nurses are influenced by both intrinsic and extrinsic factors such as time pressures, lack of motivation, long working hours, heavy workload, and stress [7,9,27,28,29,30,31,32,33,34,35,36]. Most published studies have investigated only the barriers to nurses’ practice of HPBs, using questionnaire surveys. No previously published study has used qualitative approaches to explore the enablers and barriers to Nigerian nurses’ HPBs, or organized the determinants in the context of a theoretical model which reflects multiple levels of influence within the socio-ecological environment.

Furthermore, studying both the facilitators and barriers to nurses’ HPBs ensures that interventions for improvement and sustenance of healthy habits can be implemented. Since nurses have health promotion as a core aspect of their role, and there is potential for role modeling of lifestyle behaviors to patients and the public, exploring the determinants of Nigeria nurses’ HPBs and the spheres at which these influences occur is timely and worthwhile.

The socio-ecological model (SEM) is a framework that is theory-based and enhances the understanding of the interactions between multifaceted aspects of the individual and the environmental [37,38]. SEM is underpinned on the idea that one’s behaviors influence and are influenced by their social and ecological environment. The SEM also emphasizes five hierarchical levels that are nested within each other (intrapersonal, interpersonal, community, organizational, and public policy). Intrapersonal factors are factors that directly influence individual behaviors such as attitudes, knowledge, and personality. Interpersonal factors include social interactions and its influences on an individual’s health behaviors. Organizational factors comprise the influence of structures, practices, policies, and norms within an organization on an individual. Community factors entail influences from within the wider society, which affects individuals, groups and organizations. Public policy factors comprise of national, state, and local laws and policies that influence individual’s health actions by regulation or support. These dynamic multilevel interactions are believed to be the determinants of health-related behavior. The SEM also aids in the identification of leverage points at multiple levels within the socio-ecological environment where health promotion interventions may be targeted to achieve effective impact. In our study, the SEM was useful for explaining and gaining an understanding of the determinants of individuals’ HPBs. Mapping the facilitators and barriers to nurses’ HPBs unto the SEM provided a holistic understanding of the aspect of nurses’ socio-ecological environment from which the influence arises. Furthermore, applying the SEM to this study provides information and guidance for development of single-level or multilevel health promotion interventions and programs for nurses in the future. Therefore, our study aimed to explore the perceived facilitators and barriers to engagement in HPBs among Nigeria nurses using the SEM approach.

## 2. Materials and Methods

Study design: This study aimed to understand and describe the perceived facilitators and barriers to Nigeria nurses’ engagement in HPBs based on the SEM. Therefore, a generic qualitative research design was adopted in this study. Data were collected qualitatively by means of semi-structured interviews intended to gather comprehensive and contextual information based on nurses’ perspectives. Interviews were carried out on a face-to-face, one-to-one basis with the guidance of a pre-determined interview guide with open ended questions and prompts (Appendix A). The use of semi-structured interviews afforded the production of socially relevant knowledge on factors influencing nurses’ lifestyles in a pragmatic way [39]. The interview guide for this study was informed by the literature and developed by the project team, with the intention of obtaining insights into the enablers and barriers to Nigeria nurses’ dietary, physical activity, alcohol consumption, and smoking behaviors, respectively, both at work and out of working hours (Appendix A). Interview questions were used as a guiding framework, but were adapted by the interviewer, using prompts applicable to each respondent’s responses, in order to elicit relevant information on the subject matter. The introductory section of the interviews explored nurses’ lifestyles regarding the respective HPBs. Subsequent sections distinctly addressed the facilitators and barriers influencing nurses’ engagement in respective HBPs. The nature of influence exerted by these determinants was also explored. Interviews also obtained insight into Nigeria nurses’ perceptions about interventions that may promote their engagement with healthy lifestyle behaviors, and/or eliminate or reduce perceived barriers to their regular practice of HPBs. At the end of the interviews, nurses were provided with opportunity to provide relevant additional information. Ethical approval was obtained from the University of Nottingham Faculty of Medicine and Health Sciences Research Ethics Committee, and from the Hospital Research Ethics Committee (HREC) in Nigeria prior to commencing the research study (Ref: OVS26082016). The study was carried out in a large sub-urban tertiary health facility in Nigeria. All nurses working in the medical section were invited to participate in the interviews. Expression of interest slips were distributed to these nurses, along with other information documents about the research study via their letterboxes. Interested nurses could indicate their interest by inserting completed expression of interest slips into a sealed letterbox that was placed in the nurses’ lunch room. To promote compliance and increase response rate, posters were placed in the nurses’ common rooms. Nurses who indicated interest in the study by returning their filled forms were contacted using their preferred approach at a mutually convenient time. Prior to the interviews, the researcher ensured all participants had received, read, and understood the participant information sheet given to them as well as the details of the research and that the interviews would be audio-recorded, anonymized, and confidential. Further clarifications were given by the researcher to additional questions asked. Each nurse who expressed willingness and consent to participate in the interviews filled out the informed consent form and signed accordingly. All interviews were conducted in English language, which is the official language for health professionals in Nigeria. A sample size was not pre-determined for the research study. Interviews continued until the point of data saturation, a point at which no new information seemed to emerge from further conversations [40]. Around the fifteenth interview, the researcher recognized patterns among the respondents’ experiences and further interviews confirmed the saturation of knowledge. A total of eighteen interviews were conducted for the study in total (Table 1) and the duration of interviews ranged from 25 to 65 min, although the average duration was 40 min. All audio-recorded interview data were transferred and saved on a computer, in a password-protected file with a unique identifier and pseudonym. All audio recorded clips of the interviews were then transcribed verbatim by the researcher. Interview data transcript organization and comprehensive analysis was performed with the use of NVivo10^®^ software. Guided by the pre-determined research objectives, data were analyzed using thematic analysis [41]. After familiarization with the interview data on facilitators and barriers to nurses’ engagement in HPBs, initial codes were generated through reading and re-reading of the transcripts. The initial codes generated were further analyzed, the concepts developed and then the categories formed. The categories were differentiated by being either perceived facilitators or barriers to Nigerian nurses’ engagement in HPBs. Tentative themes were generated after analyzing and grouping related categories. The tentative themes were then revised and reviewed by discussion among study authors, which led to the finalization of the themes of the study which best explained the research aims. The study authors (C.U., H.B., and R.W.) ensured that all emerged themes accurately answered the research question and adequately explored the data. The data analysis procedure was solely deductive and nine themes emerged from the data. Findings from the data analysis showed that Nigerian nurses’ HPBs are influenced by facilitators and barriers arising from the five levels of influence of the SEM (Table 2). The SEM was utilized at the later stage of the data analysis to map the themes that emerged from the data onto their corresponding levels of influence.

## 3. Results

### 3.1. Participants’ Characteristics

The interview participants were mostly females (Table 1). This reflects the gender distribution of females to males in the wider population of nurses (over 90% of nurses working in Nigeria are females). Their ages ranged from the age group 20–30 to over 50 years, with majority of the participants being between 31 and 50 years. This age distribution broadly aligns with the age bracket for the majority of the nursing workforce in Nigeria. Most of the nurses had 15 years of nursing experience or less. Bachelor of Nursing was the highest educational qualification of the nurses interviewed. Most of the nurses had the basic nursing diploma level training, which in the Nigerian context enables them to be registered nurses. In relation to self-reported health, the majority of the nurses reported their health to be either ‘very good’ or ‘good’. None of them reported being in poor health although two nurses reported their health condition as ‘fair’ while three nurses reported theirs as ‘excellent’.

### 3.2. Research Findings

The research findings reveal that Nigeria nurses’ HPBs are affected by perceived influences at all five levels of the SEM. However, in almost all the research themes, nurses reported more perceived barriers than perceived facilitators for their engagement in HPBs. Table 2 below shows the perceived facilitators and barriers to Nigeria nurses’ personal HPBs practice at respective levels of the SEM and highlights aspects where no facilitators were reported despite several existing barriers (Table 2). 

### 3.3. Intrapersonal Level

Intrapersonal level themes include (a) knowledge and experience, (b) perception, attitudes, and beliefs, and (c) personal demographics.

Nurses expressed that their health-related knowledge served as a facilitator for their practice of HPBs. The knowledge of the health benefits and consequences of unhealthy lifestyle behaviors seemed to be a further motivating influence for nurses to make healthy lifestyle choices. Nurses’ awareness of their family healthy history also exerted a positive health influence on their HPBs.


*I know that living a healthy lifestyle (…) are good for the body to be healthy. [Nurse5].*



*(…) knowing what [illness] my mum and grandma went through makes me strive for a healthier me [Nurse18].*


Although the knowledge nurses possessed did not automatically translate into engagement in healthy behaviors, especially in healthy eating, and physical activity, it regularly awakened the nurses’ consciousness to endeavor to make healthier choices, irrespective of whether they acted on their intentions. For some nurses, personally experiencing certain illnesses and symptoms directly related to, or as the consequence of, unhealthy behaviors was perceived to have an enabling influence on their current practice of HPBs. This is despite the fact that the majority of these nurses self-reported their personal health as being either good or fair. The experience of caring for people with illnesses associated with unhealthy lifestyles, was also frequently cited by several nurses as a reason to reflect on their own HPBs with consideration towards making better, healthier lifestyle decisions. Additionally, some nurses felt that their experiences of caring for patients with lifestyle related conditions mirrored the probable later consequences of their own unhealthy behaviors and seemed to put their own actions and choices into perspective.


*“I once cared for a friend who had cancer […] since then I decided to put my daily choices in check [Nurse10]”.*


Lack of knowledge about national healthy lifestyle guidelines was a significant barrier for nurses; although nurses expressed having knowledge and experience, they actually made no reference to any particular evidence-based health guideline or recommendation (in relation to healthy eating, physical activity, alcohol consumption, or smoking). Nurses mostly emphasized in general terms the importance of a nutritious well-balanced diet, exercise, not smoking, and consuming little or no alcohol. For example, nurses appeared to be unaware of the World Health Organization (WHO) physical activity guidelines, and so it was impossible for them to know whether they actually met physical activity recommendations themselves [42]. The nurses in our study were not able to adequately classify their HPBs as healthy or unhealthy based on evidence-based standards. This prevented them from assessing their own current behaviors, or engaging in decisions as to whether they needed to make improvements (or not) where necessary.


*I’m not aware of any guidelines for health promoting behaviors [Nurse8].*


Nurses’ perceptions, attitudes and beliefs contributed to determining their HPBs. As a result of their perceived value for personal health, some of the nurses reported making efforts to eat healthy foods and perform physical activity, but they acknowledged that these actions were not undertaken on a regular basis. Of nurses who described that they valued their health, a similarity between them was that they reported having one health challenge or another that made them more careful about their health and lifestyle choices.


*My health is my wealth and priority now, (…), as I have some health issues [Nurse18].*


For the nurses who described valuing their health even in the absence of any reported illnesses, being healthy and fit was their motivator and underpinning for making healthy choices especially with regards to alcohol and smoking behaviors. Indeed, smoking and alcohol behaviors were more likely to be discussed in the context of value for health compared to their dietary and physical activity behaviors.


*It all comes to how much regard I have for my health (…) I never indulge for my health sake [Nurse3].*


Positive self-regulatory mechanisms also helped nurses maintain healthy habits through the use of self-efficacy, self-discipline, and self-motivation. Nurses’ intrinsic regulatory measures were greatly enhanced by their perceived health-related knowledge. Nurses also acknowledged their challenging journeys towards adopting and maintaining healthy eating and physical activity. Nevertheless, nurses who reported being resilient about consistently being able to make healthy choices expressed general assertiveness and appeared to have confidence and hold beliefs about their capability for success with regards to long-term health.


*I push myself to eat the right food and not junk food…. I’m determined to hang on for the long term [Nurse15].*


Few nurses commented on non-health related experiences that prompt them to engage in healthy eating and physical activity. These nurses reported that their experiences were related to their negative self-image and self-perception and led to positive change towards their engagement in HPBs *[Nurse4, Nurse7, and Nurse13].*

On the other hand, a lack of self-regulatory mechanisms were expressed by nurses as a barrier to their practice of HPBs. Self-efficacy concerns were highlighted in the nurses’ inability to cope with maintaining their perceived HPBs because they found it challenging. The trend of vacillation especially in diet and physical activity behaviors was obvious from the nurses’ accounts. Nurses recalled several personal strategies, such as diet and activity plans, which they had utilized previously in order to stay active and eat healthy. However, loss of intrinsic motivation to persist and sustain the healthy behaviors was of primary concern to the nurses.


*My motivation is quite low because each time I set out to be healthy, (…) but in a few days I’m back to where I started. [Nurse14].*


A general lack of interest and concern for healthy lifestyles was expressed by some of the nurses. The indifference reported was often associated with their perceptions of healthy lifestyle activities as being an additional stress to their already busy lives. Some nurses also reported that they felt they already had good health and were fit for their job performance irrespective of their own behaviors and choices. They acknowledged healthy lifestyles are beneficial but exhibited nonchalance and a general lack of enthusiasm to go beyond their comfort zones and make any conscious effort to engage in HPBs they were not practicing.


*I’m not keen on all the details and prescriptions [for healthy lifestyles] (…) long as I get by every day, that’s ok. [Nurse2].*


Furthermore, nurses regularly mentioned time constraints and competing priorities, particularly as barriers to healthy eating and physical activities. Time constraint barriers were reported more frequently by female nurses, who were married with children, than by the single female nurses, or male nurses. Time pressure was mostly reported as a key factor influencing their unhealthy dietary choices such as unhealthy snacking because nurses perceived that they had less time for healthy meal shopping and preparation. Regarding physical activity, nurses also attributed their inability to exercise both at home and at gyms to a lack of time. Even when off work, nurses still reported having other more important activities and domestic engagements that competed for their time which they prioritized over physical activity.


*On most days, I don’t really have much time […], I might just have another fast food meal [Nurse1];*



*I don’t seem to be able to find the time to be active enough [Nurse7].*


Lack of planning was cited as a factor preventing nurses from being able to incorporate healthy eating and physical activity into their daily schedules and this was closely linked to time management concerns. Nurses perceived that not having plans resulted in them spending more time on some activities at the expense of others. In addition, the nurses also perceived that they had less choice over the foods they consumed due to poor planning. None of the nurses alluded to any sort of schedule for healthy eating and physical activity being built into their daily routines.


*I don’t plan, no plans for exercising and to cook too so I survive on snacks or fast foods [Nurse14].*


Personal preferences and habits such as comfort eating and affinity for high-fat and high-sugar foods were described by most of the nurses, and these actions were seen to inhibit their practice of HPBs because they were established habits that were seen to be difficult to break, preventing them from adopting healthier options. Taste seemed to be the main driver of meal selection, although the meal choices reported were found to consist mainly of unhealthy high-fat and high-sugar food options.


*I don’t feel that I’m depriving myself or selecting this food and rejecting that food when I eat what I want to eat [Nurse2].*


Sleep deprivation, occupational stress, and fatigue were mentioned by most of the nurses. Nurses reported experiencing disrupted circadian rhythms, and reported high daily stress levels. Increased stress levels meant that nurses’ willingness to be active or make healthy meal choices was reduced. Tiredness and stress were considered to be related to nurses needing more rest to conserve their energy for future tasks. Therefore, plans and tasks related to physical activity and preparation of healthy food were frequently put aside. The nurses also related stress to their habits of comfort eating, overeating, and unhealthy snacking, which they perceived to be mechanisms for coping with unpleasant events. The experience of stress, fatigue, and sleep deprivation was reported more among nurses who had family and dependents.


*Whenever I’m stressed, I can’t make sound decisions for my health, (…) It makes me reach for the wrong foods I would normally walk past […]. When I could have walked some distance, I find that I’m picking a taxi so I can get home and into my bed [Nurse9].*


Personal demographics of the nurses such as age, gender, religion, and socio-economic status were also found to influence their HPBs. While older nurses reported that advancing age makes them more conscious about their dietary intake due to age-related changes in their body metabolism, their engagement in physical activities greatly reduced with increasing age. The fear of injury was also expressed by older nurses as contributor to their non-engagement in regular physical activity behaviors. These findings were mostly expressed by both male and female nurses who were middle aged or closer to retirement.


*I have to be careful and mindful about what I’m eating now that I’m getting older. [Nurse11].*



*After ten minutes of exercise (…), I began to pant like I was going to pass out. I’ve never tried it [exercise] again since then. [Nurse18].*


Religious beliefs and inclinations underpinned nurses’ non-smoking and non-alcohol consumption behaviors. Almost all nurses reported being non-smokers and the vast majority stated that they consumed no alcohol. There seemed to be strong control of smoking and alcohol behaviors by religious tenets among the nurses, irrespective of the religion they belonged. A similar trend was not reported for unhealthy eating and sedentary behaviors as nurses did not link their actions in these behaviors to their religion.


*I’m not allowed by religious principles to smoke or drink alcohol, so I don’t [smoke or drink alcohol] [Nurse12].*


Male nurses explained that the nature of male gender roles meant that men were exempt from most of the domestic chores. This gave males more free time at home to fit exercises into their daily activities. Also, having wives or other females at home to prepare meals also meant that planning and preparation of meals was not a concern for male nurses, leaving them less time constrained. Female nurses perceived that their cases were the exact opposite of that of the males. They explained that having several domestic chores and family roles in addition to nursing work bars them from regular engagement in physical activity and healthy eating. Female gender roles were very frequently associated with time constraints and increased workload as nurses considered the summative influence of their professional and domestic work demands on their HPBs.


*I’m busy round the clock, wearing different hats…a fulltime nurse, a mum, a wife, a daughter, a sister…all these hats carry responsibilities, it’s a lot to take on [Nurse13].*


Several nurses explained that having low socio-economic status prevented them from affording the cost of regular healthy eating, especially the high cost of fruits and vegetables for the family and electricity for food storage. Affordability of gym subscriptions and exercise equipment was also reported as a negating factor to their practice of physical activity. Nurses linked their inability to afford their perceived cost of healthy eating and physical activity to low income, in the face of other competing financial demands and household bills to be paid.


*I don’t earn enough to afford eating well [healthy meals] every day (…). It’s even more when you add the money for family care, transport, bills, and school fees…oh no! I can’t afford that [healthy meals] [Nurse17].*


### 3.4. Interpersonal Level

At the interpersonal level, social media, and relationships nurses have within their social circle were found to influences their health promoting behaviors in both healthy and unhealthy ways. Internet and social media platforms represented useful information resources for healthy lifestyles which nurses accessed regularly. Recipes for meals and different types of exercises were among the lifestyle resources accessed online though only a few nurses, mostly younger nurses, mentioned utilization of the internet. These nurses expressed that they derive motivation for healthy behavior from online information and interactions.


*I browse and see people eating healthy and being active (…) that’s very encouraging. [Nurse10].*



*(…) So many nice healthy foods there (Facebook group) and simple exercises you can do at home. I’ve downloaded them to use. [Nurse6].*


Family traditions and value systems appeared to be a strong motivator for enabling nurses to abstain from smoking and risky alcohol consumption. Most nurses recalled the unacceptability of smoking and alcohol consumption in their families, especially among females. Nevertheless, family influence did not appear to have a similar enabling influence on the nurses’ dietary and physical activity behaviors. Conversely, the nurses perceived that their social circle had a negative influence on their dietary and physical activity behaviors. For most nurses, their family, peers and colleagues were perceived to make healthy lifestyle practice challenging by further reinforcing and encouraging decisions related to unhealthy eating and physical inactivity; social pressures were highly prevalent at home and at work. Lack of support, persistent enticement to indulge in unhealthy behaviors, and pressure to conform to group norms were the main challenges at the interpersonal level of influence, preventing nurses from healthy eating and physical activity engagement.


*When you want to be healthy, they just laugh and say ‘yeah, we know you…don’t even bother (…), you can’t do this’…that’s so discouraging. It’s hard to keep up [with healthy eating and physical activity] when I don’t feel supported by either my family or my friends [Nurse3].*



*Regular snacking is our usual get-away, it’s like a tradition among us [nursing colleagues] [Nurse12].*


### 3.5. Community Level

At the community level of influence, the physical environment was found to negatively influence nurses’ engagement in HPBs. Nurses reported lack of infrastructure and felt that the built environment and infrastructure within the community was unsupportive of an active lifestyle, this concern prevented performance of outdoor exercise, such as active travel—walking or cycling to work—or outdoor recreational physical activities or sports. Healthy food and exercise equipment were not easily accessible and affordable to nurses within the society. Additionally, preservation of healthy food items such as fruits and vegetables were challenging for nurses, leading to wastage due to lack of refrigeration and poor electricity supply. On the contrary, the community was reported to consist of a multiplicity of fast food outfits and food shops that sold predominantly calorie dense, high-fat and high-sugar snacks, food, and drinks.


*There are no good roads, no sidewalk…The electricity conditions here too (…) it’s very poor so I can’t preserve the perishable foods (…), but there are lots of restaurants and shops everywhere tempting you to come and buy junk. [Nurse15].*


The weather condition in Nigeria for most months of the year was also reported as unsuitable for engaging in outdoor physical activity due to the hot temperatures. Unfavorable weather temperatures result in discomfort and an unpleasant physical activity experiences for nurses, thus increasing their likelihood to quit their practice of regular physical activity.


*I don’t like the sweaty sticky feeling of exercising in the hot weather [Nurse9].*


Nurses expressed concern for the unhygienic condition of the towns, poor traffic control, and security situation within the community. They perceived that they were limited in the times and places they commute. These issues further compounded their ability and interest in mobility, and also the regularity with which they have healthy meals at home and engage in activities.


*Imagine I went out to jog, I’ll be at risk of being knocked down [Nurse15].*



*There has been too much kidnapping recently, even among nurses [Nurse14].*


Negative cultural beliefs and social norms around involvement in smoking and alcohol behaviors were perceived by nurses to have a positive impact on the respective behaviors. These behaviors were reported to be culturally and socially unacceptable in Nigeria, especially among females, though were still practiced by some. The social identity of nurses also meant it was improper to be seen smoking or consuming alcohol, especially when wearing the nurses’ uniform.

On the other hand, unhealthy eating behaviors and physical inactivity were reported to have widespread acceptability within cultures and societies in Nigeria. The prevailing social acceptability of unhealthy eating, physical inactivity, and obesity indicated that people were celebrated for their weight gain and sedentary lifestyles. From the nurses’ accounts, being overweight and obesity were perceived positively within the culture, while slimmer body frames were perceived as indications of malnutrition and need for diet fortification. Traditional rites around marriage, pregnancy, and childbirth also contributed to unhealthy eating and physical inactivity among women. Certain perceptions and stereotypes within the society prevented nurses from being physically active and eating healthy meals. For example, walking and cycling were perceived to be reserved for the financially challenged. Perceived stigmatization associated with slender body size was also reported to be a barrier to physical activity, as nurses who were more active and/or lost weight received negative comments from their peers.


*When I decided to make healthier choices (…), [people] were asking me what the matter was that was eating me up, because I lost some weight (…) some even demanded that I gained back the weight [Nurse18].*


Nurses pointed out that the media worked contrary to their healthy eating intentions. The media was portrayed by nurses to be an embodiment of adverts with continual enticements to indulge in certain high-fat and sugary foods; these were irresistible for nurses and undermined their self-control. Adverts were perceived to display more unhealthy foods and very few healthy meals.


*It’s hard to look away, those picture-perfect adverts of the unhealthiest foods are all around, so many promos and free gifts too, they never do that with fruits [Nurse9].*


### 3.6. Organizational Level

Factors related to nursing practice and hospital management constituted organizational influences on nurses HPBs. Nurses reported that getting days off from work meant that, in theory, they could use this time to recuperate and engage in healthy meal preparation and physical activity. The nature of nurses shift work was such that nurses work between 8 to 12 h shifts for about four to five day or night shifts and get about two to three days off work. Days-off were the only facilitator for nurses at the organizational level of the SEM.


*My off days are my active days at home, I always look forward to it [days off], to relax well, sleep, shop, and cook some nourishing food. [Nurse6].*


For most of the nurses, especially the female nurses, working in a rotating shift work of doing night and day shifts interchangeably was a strong barrier to HPBs. The nurses strongly linked shift work to barriers experienced at the interpersonal level especially sleep deprivation, lack of time for planning and perceived time constraints as described above. The nurses reported experiencing negative health consequences of shift work, and believed that shift work was a factor that prevented them from engaging in healthy eating and physical activity.


*Working mornings, evenings, and nights irregularly mean I don’t have a routine (...), it’s not good for my body [Nurse10].*


Nursing as an occupation was associated with heavy workload, and this was linked specifically to the geographical setting, since nurses described high patient coverage within the Nigerian healthcare facility. Nursing shortage was reported by all nurses to be the main cause of their heavy workload. Being busy at work was associated with a sense of time pressure during the working day which affected nurses getting hydration and regular meals. Busy schedules for the nurses often resulted in meal skipping or having irregular meal schedules, unhealthy snacking, and then overeating when they had gone long periods without food, which caused them to feel stressed and led to fatigue.


*Being short-staffed means we have lots of work to do…you don’t think of yourself at those times (…), there’s hardly time to eat, drink, or urinate. [Nurse12].*


Lack of scheduled on-shift breaks also had significant negative impact on nurses’ HPBs. The nature of Nigerian nurses’ shift schedules were reported to involve no scheduled on-shift breaks. The lack of protected breaks was seen to be a major barrier to healthy eating, and to engaging in physical activity. The personal stress caused by a lack of breaks was viewed to be a contributing factor to poor choices, particularly with regards to diet. All of the nurses felt that having protected breaks would alleviate stress, and would likely lead to changes in their habits, such as taking walks, resting, and thinking more carefully about food intake and food choices.


*I don’t get any breaks, […] we have to eat handy snacks on the go or forget about eating until after handover, that is not helpful [Nurse6].*


Hospital management and environmental barriers included inadequate infrastructure, poor access, high-cost and low-quality of hospital food, and perceived employer lack of value for nurses’ health. Most of the nurses commented on the inadequacy of facilities for healthcare workers within the hospital. Nurses reported a lack of adequate space to relax and recuperate away from patients. This concern was associated with increasing nurses stress levels. A recurring challenge was that of poor electricity supply within the hospital, which indicates that the hospital environment posed similar barriers as the community environment.

From the nurses’ accounts, the hospital environment does not support healthy eating. Lack of breaks and adequate food storage facilities, coupled with working night shifts was reported to make access to food difficult for nurses, thereby promoting unhealthy snacking behaviors. Within the hospital, unhealthy foods and snacks were the more convenient and accessible options, offered at cheaper prices. Snacks and foods available to purchase within the hospital were reported to be of low nutritional quality. The nurses perceived that their employing organization did not value their health and wellbeing. This perceived lack of value for nurses’ health by hospital management was associated with the absence of any facilities or services to support their wellbeing, such as healthy food outlets or workplace health initiatives, while in contrast, they spoke of the presence of multiple triggers for unhealthy dietary behaviors and physical inactivity.


*It’s hard to find nicely prepared healthy food to eat around here [hospital environment], it’s either junk food, too far to get to or sold for very high price. It’s even harder on night shift when all restaurants have closed, [Nurse3].*



*(…) we [nurses] should be looked after so we can look after the patients well [Nurse13].*


### 3.7. Public Policy Level

Concerns around government leadership and policy were reported as barriers to HPBs among nurses. There was no facilitator for nurses’ practice of HPBs reported for this level of influence on the SEM. Nurses perceived a national lack of concern and prioritization for public health and linked this concern to the government of Nigeria not placing adequate value on the health of the citizens more generally. This perception was related to the lack of adequate infrastructural development to support healthy lifestyle choices, and a lack of preventive interventions and an absence of health promotion programs more broadly, across the country. Nurses also felt that the government was responsible for dissemination and publicizing healthy lifestyle guidelines and recommendations nationally through the Ministry of Health and other sources to create awareness among its citizenry, but that this was inadequate on a national level.


*Nothing is ongoing to improve our health and lifestyles; the government is not doing enough [Nurse11].*


Nurses also reported the government’s inadequate implementation, regulation, and monitoring of health promotion policies. Some of the nurses commented on the government failing to implement bills related to improving the living standard of workers in order to ameliorate financial challenges to healthy eating. Nurses explained that healthy food options such as fruits and vegetables are more expensive to purchase than unhealthy high-carbohydrate, sugary and fatty foods.


*The government and labor unions need to implement our pay rise bill…so we can afford healthy diets every day [Nurse6].*


It was generally viewed that the government did not appropriately regulate or monitor interventions for the promotion of healthy lifestyles. For example, media advertisements were seen to advocate unhealthy lifestyles, and nurses proposed that the multiplicity of unhealthy foods and promotions needed to be cautioned by the appropriate bodies within the government.


*If the government were sensitive to these junk food adverts, they would see that they are not helping people (…). The government is doing nothing about that [Nurse5].*


## 4. Discussion

This study aimed to explore the perceived facilitators and barriers to Nigerian nurses’ engagement in HPBs using the SEM. Study findings show that Nigerian nurses’ practice of healthy dietary and physical activity behaviors is affected by more perceived barriers than facilitators at all levels of SEM. The determinants of Nigerian nurses’ HPBs are also clustered at the intrapersonal level and increased perception of barriers to healthy eating and physical activity is associated with decreased engagement in these behaviors [43,44]. On the other hand, strong religious, social, and socio-cultural factors underpinned non-smoking and low-risk alcohol consumption behaviors among the nurses. Previous research found that religion exerted a strong deterrent effect on smoking behaviors [45] and was associated with less alcohol use, especially among women [46].

Furthermore, most of the constructs such as knowledge, perceptions and attitude, and personal demographics influenced nurses’ dietary and physical activity behaviors, and these health behaviors seemed to cluster together [47,48]. Therefore, interventions to sustain facilitators and address the barriers to healthy dietary behaviors may also result in improvement in physical activity performance among Nigerian nurses [49]. Practically, health promotion interventions that target related lifestyle behaviors such as energy-balance interventions to improve food intake and maintain a healthy weight, alongside physical activity for energy expenditure should be introduced to the Nigerian nursing population. Given that HPBs result from complex interaction of multi-level behavioral determinants, positive changes in health-related behaviors are more likely to last when the change simultaneously targets the individual and their environment [37].

Nurses as health professionals may be expected to practice healthy lifestyles and be role models for health [3,50,51]. Nevertheless, the inconsistency in the literature about the mixed influence of knowledge on HPBs may be explained by the relationship between psychological factors, such as self-regulatory mechanisms, and behavior. Consistent with our findings, previous studies have also reported that nurses’ health related knowledge may not directly translate into healthy diet consumption and physically active lifestyles [3,6,7,9,35,43,52,53]. Additionally, despite their occupation and prior training, many of the Nigerian nurses in the sample had limited knowledge about national health-related guidelines and so there is scope for development of educational interventions for nurses around health behavior change. The health promotion content of nursing education in Nigeria warrants further investigation as there may be scope for educational interventions specifically targeting nurses-in-training as the next generation of healthcare workers.

In line with interpersonal determinants found from this study, other studies have highlighted the mixed influence of family and social support on HPBs [32,44,53,54,55,56,57]. Specifically, nurses have been found to have both negative and positive influences on the HPBs of their work colleagues, [31,36] yet our study found that Nigerian nurses influenced only unhealthy behaviors (such as unhealthy diet, poor food choice, and physical inactivity) among their work colleagues. Social norms around eating behaviors and physical activity clearly determined nurses’ engagement (or not) with these respective behaviors, and previous research also reported similar findings [58]. This demonstrates a need to consider efforts to change health-related culture and attitudes within this working environment, as well as more widely in Nigerian society.

At the organization and public policy levels of the SEM, there were a lack of facilitators for nurses to engage in healthy eating and physical activity. Although organization and policy level barriers seemed to be beyond the nurses’ sphere of control, nurses are constantly affected, and most impacts were experienced at the intrapersonal level. Nurses’ self-regulatory mechanisms and self-efficacy may also have become undermined in the presence of several higher level barriers. It is important for organizations to recognize their responsibility towards supporting employees in the uptake and maintenance of healthy behaviors [59]. In the Nigerian healthcare context, there was a lack of education, or intervention in the areas of self-care or workplace health and wellbeing. The lack of learning packages within nurses’ continuing professional development (CPD) training, which are targeted at the modification of lifestyle behaviors, has been highlighted by previously published studies in the United Kingdom, United States of America, and Australia [11,60,61]. Nigerian nurses could benefit from tailored workplace health promotion interventions promoting healthy eating and physical activity, as have been delivered in other geographical regions [62,63]. Health promotion interventions may help to build nurses’ self-efficacy for healthy eating and active lifestyles; also, interventions to build resilience may alter their perception of barriers and their influence (such as to reduce the negative impact of behavioral ‘norms’ instilled by work colleagues). This might help nurses to be more confident to change maladaptive personal health beliefs that appear to exist in the presence, or absence of good intentions to act, and overcome barriers to change. In a study by Jackson et al., [64], the value placed on health and health-related self-efficacy was associated with engagement in HPBs and predicted the levels at which people engage in healthy lifestyles.

In Nigerian healthcare settings, nurses practice rotational shift work of about 8 to 12 h per shift, on a full-time basis. This working pattern is similar to that of nurses in developed countries and is aimed at providing 24-hour healthcare services despite acute nursing shortages, although the direct results for individual employees is busy shifts, heavy workload, and long working hours. These organizational barriers are also associated with several individual level barriers such as disrupted circadian rhythm and adverse health effects [65,66,67]. Work-related stress in nurses has also been reported in studies of nurses in a range of international contexts [12,32,34,35,36,68,69,70,71,72]. Stress results from unfavorable working conditions, such as lack of scheduled on-shift breaks, and negatively affects nurses’ likelihood to engage with HPBs. In the long term, working in an unhealthy reality predisposes nurses to chronic stress, further adoption of unhealthy behaviors, increased body mass index (BMI) [27,73,74,75], as well as higher risk of chronic diseases [76]. A study of the stress experience of Nigerian nurses found that vast majority of the nurses reported perceived stressed which negatively affected the nurses’ HPBs [77]. Interestingly, nurses have also been found to build resilience by engaging in healthy lifestyle behaviors as a positive coping mechanism for work-related stress [9,31]. Nevertheless, development of strategies to improve the health and wellbeing of nurses and other hospital employees lies with the employing organizations [78]. Previous studies have shown that workplace health promotion programs (WHPP) hold benefits for improving employee HPBs [79,80,81], and therefore we recommend that WHPP interventions are offered to the workforce in Nigerian hospitals.

At the policy level, nurses perceived that the government needed to prioritize preventive health strategies on a national level. Interventions to promote the practice of HPBs in the general population are lacking and urgently warranted. Therefore, we advocate the implementation of policies and best practices that expand choices and options for healthy lifestyles within society more broadly, as well as the review and modification of the cost associated with healthy foods and physical activity options. Policies for monitoring, and regulation or restriction of public promotion of unhealthy lifestyles [82] may also reduce nurses’ perceived barriers. Mass media campaigns may represent useful strategies to promote healthy lifestyles [83,84,85]. Although our study may not directly influence policy or change existing cultures of laissez-faire governance in Nigeria, it forms the basis for future research on policy issues around healthy lifestyles for Nigerians in general. We suggest that health policies consider initiation of interventions (both nationally, and within workplace settings) that are coordinated in ways that increase their potential to change ingrained behaviors, beliefs, and perceptions of nurses and Nigerian citizens. This may be achieved through a combination of social change, habit formation, and knowledge improvement [86]. The cost-effectiveness of programs aimed at improving HPBs needs to be established to determine the potential for long-term benefits in supporting behavior change. Therefore, researchers might focus on identifying the optimal designs for behavior change initiatives that are essentially low cost, but have sustained effects. Deeply grounded social and cultural norms affecting HPBs may also need to be tackled at the policy level by behaviorally informed policies. Availability of local data ensures that collaboration between researchers, policy makers, and health practitioners would enhance effective translation of health research into practice and policy development [87].

This is the first study to explore the determinants of Nigerian nurses, HPBs using the SEM. The use of semi-structured, one-to-one interviews allowed for the collection of rich data without limiting nurses to a set of pre-determined options that may not apply to their perceived HPBs determinants. The use of a saturation point to determine when to conclude data collection meant our coverage of the subject matter was appreciable.

Our study has implications for research and practice. Findings from this study add to a limited evidence-base on Nigerian nurses’ health related behaviors. Details of the enablers and barriers to nurses HPBs can potentially inform planning of strategies for healthy lifestyle education and promotion interventions within Nigerian healthcare organizations and higher education settings. Intrapersonal barriers such as lack of motivation and self-efficacy, time constraints, and lack of planning may be addressed at the organizational level through education, training, and personal development conferences. This may include information on government recommendations and guidelines for health behaviors and healthy weight, options for quicker and less expensive healthy meals and snacks, and examples of incidental physical activities that are safe, require no equipment and incur no financial cost. Suggestions for healthy dietary and physical activity behaviors may also be tailored to suit nurses’ personal preferences and lifestyles. Nevertheless, behavior change interventions should motivate nurses to prioritize their health not just because of the increased health risk from non-engagement in HPBs, but to improve their perceptions and beliefs about healthy eating and physical activity [43]. Since nurses play a key role in health promoting with patients, a shift in attitude and social norms seems paramount, not only to improve their own health and wellbeing, but to enhance the quality of patient care (with regards health promotion practice) and therefore contribute to improvements in Nigeria’s population health.

Our study is not without limitations, due to budget constraints, we recruited nurses from one healthcare facility, making our study sample homogenous. Interviewing nurses from other regions, other hospitals or other sectors (e.g., private sector) may have provided a broader understanding of the subject matter and allow for sector or geographical comparison. Nevertheless, this research is a significant addition to the evidence-base since there are currently very few studies in this field from Nigeria or Africa. This study highlights an urgent need for research and intervention in this region.

## 5. Conclusions

This is the first study to provide insight into the facilitators and barriers to Nigeria nurses’ engagement in HPBs with regards dietary habits, physical activity, smoking, and alcohol consumption. Our study findings are unique and represent Nigerian nurses’ current circumstances given that socio-economic, geo-cultural, and contextual differences make findings from studies on nurses in developed countries directly inapplicable to the context of Nigerian nurses. Findings have been mapped onto the SEM, with obesity, unhealthy dietary habits, and physical inactivity being particular concerns in this group since socio-cultural and religious beliefs advocated abstinence with regards to smoking and alcohol. This research shows that nurses experience significantly more barriers to their practice of HPBs compared to facilitators. Nurses report a general lack of policies and interventions to support health and wellbeing, both nationally and within healthcare institutions. There were a high number of societal, organizational, and job-related barriers to healthy lifestyle behaviors, and nurses did not feel that their employer valued the wellbeing of its workforce. Many nurses lacked knowledge of national guidelines for health in these areas, and further education could be provided to Nigerian nurses to clarify the national and global recommendations for health-related behaviors. Occupational stress, high workload, lack of protected breaks, and shiftwork were particular barriers to healthy diet and active lifestyles. Interventions are warranted to support Nigerian nurses’ health and wellbeing, but these should take into consideration these job-related barriers, and be specific to the Nigerian healthcare context since workplace wellness programs from developed countries may not be appropriate in resource-poor environments. Government policies should prioritize the promotion of population health through the workplace setting, by advocating the development, implementation, regulation, and monitoring of healthy lifestyle policies.

## Figures and Tables

**Table 1 ijerph-17-01314-t001:** Demographic characteristics of participants.

Variable	Category	*n*	%
Gender	Male	2	11.1
	Female	16	88.9
Marital status	Married	12	66.7
	Single/Divorced/Separated	6	33.3
Employment status	Full-time	18	100
Ethnicity	Black Nigerian	18	100
Dependents	Yes	17	94.4
	No	1	5.6
Age	18–30	3	16.7
	31–40	7	38.9
	41–50	5	27.8
	>50	3	16.7
Self -reported health	Excellent	3	16.7
	Very good	6	33.3
	Good	7	38.9
	Fair	2	11.1
	Poor	-	-
Years of nursing experience	0–5+	3	16.7
	6–10+	5	27.7
	11–15+	4	22.2
	16–20	3	16.7
	>20	3	16.7
Nursing designation	Nursing officers	3	16.7
	Principal nursing officer	4	22.2
	Senior nursing officers	6	33.3
	Chief nursing officers	5	27.7
Shift worked	Rotating	18	100
Educational qualification	Nursing diploma	16	88.9
	Bachelor of Nursing	2	11.1

**Table 2 ijerph-17-01314-t002:** Themes for facilitators and barriers to Nigeria nurses’ engagement in health promoting behaviors (HPBs) mapped onto the five levels of the socio-ecological model (SEM).

SEM Levels	Main Theme	Categories	
		Perceived Facilitators	Perceived Barriers
Intrapersonal	Knowledge and experience	Knowledge of health promoting behaviors	Unawareness of healthy lifestyle guidelines
		Knowledge of family health history.	
		Personal experience of lifestyle related illness.	
		Experience of caring for people with lifestyle related illness	
	Perceptions, attitudes and beliefs	Perceived value for personal health.	Indifference
		Positive self-regulatory mechanisms	Lack of self-regulatory mechanisms
			Time constraint
			Lack of planning
			Personal preference and habit
			Sleep deprivation, stress and fatigue
	Personal demographics	Aging	Aging concerns
		Religion	Income
		Male gender roles	Female gender roles
Interpersonal	Social circle relationships	Internet/social media influence	Family/ Peers /colleagues influence
		Family influence	
Community	Physical Environment	*** None reported	Lack of Infrastructure
			Weather
	Sociocultural factors	Cultural beliefs and Social Norms	Cultural beliefs, social norms, and stereotypes
			Access, cost, and quality concerns.
			Hygiene, safety, and security concerns
			Media advertisements
Organizational	Nursing related factors	Days off	Shifts work
			Heavy workload
			Lack of scheduled on-shift breaks
	Hospital management and environment	*** None reported	Inadequate infrastructure
			Access, cost, and quality of hospital food.
			Perceived lack of value for staff health
Public policy	Government leadership and policy issues	*** None reported	Perceived lack of concern for health promotion
			Inadequate implementation, regulation, and monitoring

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
