# Peer review of "Perceived Facilitators and Barriers to Nigerian Nurses’ Engagement in Health Promoting Behaviors: A Socio-Ecological Model Approach"

_ijerph, 2020, doi:10.3390/ijerph17041314_

Round 1

Reviewer 1 Report

Introduction: it is necessary to clearly define the HPBs. Required to contextualize the nursing profession in the scenario that the research is taking place, and not only talk about it in general; understanding that nursing work in front of HPBs is different in primary or tertiary level.

Study design: it is essential to define the type of qualitative study: phenomenological, hermeneutical, grounded theory, or other; in coherence with the investigation. To identify the approval number of the ethics committee and the informed consent application.

Results: The tables are self-explanatory, and it is not necessary to repeat the information they contain in narrated form. the table's titles need numbering, citation in the text, and year. Conceptualize the five levels of the SEM.

Discussion: give a logical order to the discussion, it could be by levels by adding subtitles or by paragraphs.

Conclusions: It is necessary to give an answer to the objective, in addition to contextualize the issues raised in the conclusions previously.

Author Response

1st Reviewer’s comments

Authors’ responses

Introduction: It is necessary to clearly define the HPBs.

HPBs have now been clearly defined and referenced in the introduction section of our paper.

Required to contextualize the nursing profession in the scenario that the research is taking place, and not only talk about it in general; understanding that nursing work in front of HPBs is different in primary or tertiary level.

In the introduction section, we have explained nursing roles within the Nigerian context in relation to health promotion at the primary, secondary and tertiary levels.

Study design: It is essential to define the type of qualitative study: phenomenological, hermeneutical, grounded theory, or other; in coherence with the investigation.

The study design section has been revised to provide further information about the qualitative study design.

To identify the approval number of the ethics committee and the informed consent application.

The ethics approval number of our research has now been included in the study. The informed consent process has also been explained in the methods section.

Results: The tables are self-explanatory, and it is not necessary to repeat the information they contain in narrated form.

The findings table has been removed from the body of the manuscript and attached as ‘supplementary file 2’. Results section has also been revised to include a narrative explanation of the results and the context of results supported with shorter illustrative quotes.

The table's titles need numbering, citation in the text, and year.

Table title have been numbered and cited appropriately in the text.

Conceptualize the five levels of the SEM.

The socio-ecological model has been conceptualised further down the introduction section. Justification for its use has also been explained further.

Discussion: Give a logical order to the discussion, it could be by levels by adding subtitles or by paragraphs.

We have revised the discussion and feel that the ordering is now more logical. Although we have not used sub-headings, we have logically sequenced the material using paragraphs. This has helped to avoid repetitions at certain crucial points and we have holistically discussed the findings because they naturally overlap in some places.

Conclusions: It is necessary to give an answer to the objective, in addition to contextualize the issues raised in the conclusions previously.

The conclusion section has been revised to highlight specifically how the study answers the research question/objective. Findings have been summarised and contextualised in the conclusion section.

Reviewer 2 Report

The study design and argument lack a level of refinement that one might expect in a research paper. The author should clarify, especially in the Introduction section, what the particular thrust of his approach is. A serious problem with the paper is that the author doesn’t make it clear where this research stands in the literature. Data gathering and data analysis can be reconsidered and discussed more comprehensively. The author should detail how other studies develop the empirical analysis (i.e. type of data, experiment, etc.) in order to allow the reader to better compare their work with other existing studies and understand their contribution to the literature. The chosen methodology seems, at points, to be narrower than what is needed to support the broader conclusions of the work. While the line of argumentation is traceable throughout the paper, it is minimally supported and thus weakened at several crucial points. The results as they are currently presented appear superficial and more context and discussion of the results is required. Where can future research further contribute to this area and what are the possible implications of this research on literature, practice and policy? The author should work harder on the approach adopted, establish a clear theoretical background to contextualize the analysis and narrow the scope of the analysis to specific aspects. Too many old references.

Some research suggestions that may complement your approach:

Byrne, Seth (2019). “Remote Medical Monitoring and Cloud-based Internet of Things Healthcare Systems,” American Journal of Medical Research 6(2): 19–24. doi:10.22381/AJMR6220192

Krech, Sue (2019). “Medical Big Data Analytics and Smart Internet of Things-enabled Mobile-based Health Monitoring Systems,” American Journal of Medical Research 6(2): 31–36. doi:10.22381/AJMR6220194

Author Response

2nd Reviewer’s comments

Authors’ response

The study design and argument lack a level of refinement that one might expect in a research paper.

Thank you very much for your comment. The study authors have extensively reviewed and refined the paper to reflect the reviewers concerns.

The author should clarify, especially in the Introduction section, what the particular thrust of his approach is.

We have revised the introduction section and further explained the approach of our research. The methods section has also been revised to further explain our study methods.

A serious problem with the paper is that the author doesn’t make it clear where this research stands in the literature.

In the introduction section, we have clarified the findings of previous research studies on the subject matter of facilitators and barriers to nurses’ health promoting behaviours. We have also identified and highlighted the evidence gap that our study sets out to investigate.

Data gathering and data analysis can be reconsidered and discussed more comprehensively.

More explanation about the data collection and data analysis processes has been added to the methods section of the paper.

The author should detail how other studies develop the empirical analysis (i.e. type of data, experiment, etc.) in order to allow the reader to better compare their work with other existing studies and understand their contribution to the literature.

It is unclear what revisions are expected by this feedback. Our study was qualitative and lent itself to qualitative data analysis procedures. Therefore, no experiments or interventions were carried out as part of our study. As stated above, we have made revisions to the introduction, which we anticipate have addressed this comment.

The chosen methodology seems, at points, to be narrower than what is needed to support the broader conclusions of the work.

Thank you for your comment. We have discussed this within the research team and believe that the chosen methodology was broad enough to gather sufficient data to answer our stated research questions.

While the line of argumentation is traceable throughout the paper, it is minimally supported and thus weakened at several crucial points.

We have significantly revised our study manuscript and included more supporting references to support the arguments made in the research paper.

The results as they are currently presented appear superficial and more context and discussion of the results is required

The entire results section has been revised to address superficiality and more context and narratives have been included for clarity.

Where can future research further contribute to this area and what are the possible implications of this research on literature, practice and policy?

In the discussion section, we have highlighted areas where further research can contribute to the subject matter. The implications of our study for literature, practice and policy has also been featured in our study discussion.

The author should work harder on the approach adopted, establish a clear theoretical background to contextualize the analysis and narrow the scope of the analysis to specific aspects.

The study was a generic qualitative study and data analysis followed the guidelines for thematic analysis by Braun and Clark (2006). We believe that the scope of the analysis if narrowed further would not appropriately address the research aims.

Too many old references.

Although there are some older papers referenced, the points made in them are still relevant. We have conducted literature search in this area, which clearly suggests the need for more recent evidence in the area of nurses’ health and health promoting behaviours. However, we have recognised this as a limitation. Please note that there is very little, if any, evidence relating to the specific population of this study (Nigerian nurse’s health and wellbeing). 

Some research suggestions that may complement your approach:

Byrne, Seth (2019). “Remote Medical Monitoring and Cloud-based Internet of Things Healthcare Systems,” American Journal of Medical Research 6(2): 19–24. doi:10.22381/AJMR6220192

Krech, Sue (2019). “Medical Big Data Analytics and Smart Internet of Things-enabled Mobile-based Health Monitoring Systems,” American Journal of Medical Research 6(2): 31–36. doi:10.22381/AJMR6220194

Thank you for your suggestions. Due to access constraints, it was not possible for get the full-text of the articles. We have accessed the abstracts and have chosen not to include details based on this, as they are not directly related to the main subject of our paper.

Reviewer 3 Report

Comments to author

Thank you for the possibilty to read your manuscript. The manuscript concerns an interesting and important topic in the field of health promotion research. The study seems to be well performed however the method needs to be further elaborated and described in order to understand the quality of the study. Significant information is touched upon in the method section but some parts needs to be further clarified (see the following comments). The findings are interesting but some content are duplicated in several sub-categories. I believe that the paper have potential to contribute with new knowledge if the results are further developed.

Introduction

The introduction includes important and relevant literature. However, there is a lack of a clear rational for this study. It is possible to briefly understand why this study is needed but it might be clarified. At this point it is written between the lines. Please, clarify the rational with a few sentences.

The SEM is central in this study and you are touching upon the model in the introduction. However, in order to make it easier for the reader to grasp the whole study and the findings in particular it would be valuable with a more in-depth description of the model and an explanation of what it adds to your study.

Method

At page 2, line 74-75, you are writing that an invitation was sent out to 210 nurses and then only 18 participants were included. This needs to be commented on either in the method section or in the methodological discussion session. What about potential selection bias? Based on these preconditions it is also necessary to consider the use of data saturation. Is it possible to reach data saturation with 18 participants? Please comment on this in the manuscript.

There are no information regarding the interview guide. It would increase the quality of the reporting if some information regarding the guide was added. Is it possible to give some examples of questions or the structure of the guide?

The data analysis procedure is vaguely described and it is not possible to follow the steps that have been conducted. This needs to be clarified. For example, you write that recorded clips were transcribed verbatim. How was these clips selected? Furthermore, you need to develop the description on how the SEM was used in the analysis. From the beginning or in a later stage of the procedure? Is the analysis solely deductive? The description of data analysis needs to be further developed in order to really understand what you have done.

Results

Overall, the results include several important factors that are facilitators or barriers in the nurses HPB which is good and harmonize with the aim of the study. Still, the findings are really extensive and some sub-categories are overlapping each other which makes it hard to read. For example, the text on line 203-207 lies very close to the text written in the sub-category “personal experiences of life style related illness”. Another example is that time constraints are closely related and duplicated to some extent in the sub-categories “sleep deprivation, stress and fatigue” and “female gender role”. This might be due to the use of the SEM in the analysis procedure but if that is the case it needs to be clarified in the manuscript. If that not is the case I think the quality of the manuscript would improve significantly if the finding section was went through one more time and the categories and sub-categories further elaborated.

Discussion

Important aspects are brought up in the discussion and I think this part of the manuscript is well written.

In the paragraph focusing on methodological considerations it would be beneficial if you could elaborate on the issues I brought up in my comments regarding the method.

Author Response

3rd Reviewers comments

Authors’ responses

Thank you for the possibility to read your manuscript. The manuscript concerns an interesting and important topic in the field of health promotion research. The study seems to be well performed however the method needs to be further elaborated and described in order to understand the quality of the study. Significant information is touched upon in the method section but some parts needs to be further clarified (see the following comments). The findings are interesting but some content are duplicated in several sub-categories. I believe that the paper have potential to contribute with new knowledge if the results are further developed.

Thank you for your comment. The study results section has been revised and developed further to address content duplication. Sub-categories have been narratively described and findings currently reflect the determinants under the five levels of the socio-ecological model though guided by the study themes.

Introduction

The introduction includes important and relevant literature. However, there is a lack of a clear rational for this study.

It is possible to briefly understand why this study is needed but it might be clarified. At this point it is written between the lines. Please, clarify the rational with a few sentences.

A clearer study rationale has been included in the introduction section.

The SEM is central in this study and you are touching upon the model in the introduction. However, in order to make it easier for the reader to grasp the whole study and the findings in particular it would be valuable with a more in-depth description of the model and an explanation of what it adds to your study.

The SEM has been further explained/contextualised in line with the justification for use in our study.

Method

At page 2, line 74-75, you are writing that an invitation was sent out to 210 nurses and then only 18 participants were included. This needs to be commented on either in the method section or in the methodological discussion session. What about potential selection bias? Based on these preconditions it is also necessary to consider the use of data saturation. Is it possible to reach data saturation with 18 participants? Please comment on this in the manuscript.

Our study has outlined that we relied on the concept of data saturation to determine when to stop conducting further interviews as no new data emerged beyond point of data saturation.

Is it possible to reach data saturation with 18 participants? We have included explanation of data saturation as related to homogeneous population as was the case in our study of nurses working in the same department and organisation. From participants’ demographics, respondents were 100% Nigerian and almost 90% female.

There are no information regarding the interview guide. It would increase the quality of the reporting if some information regarding the guide was added. Is it possible to give some examples of questions or the structure of the guide?

The interview guide has now been provided in ‘supplementary file 1’.

The data analysis procedure is vaguely described and it is not possible to follow the steps that have been conducted. This needs to be clarified. For example, you write that recorded clips were transcribed verbatim. How was these clips selected?

We did not select clips for transcription. All interviews conducted were transcribed verbatim before analysis. The manuscript has been corrected to clarify this.

Furthermore, you need to develop the description on how the SEM was used in the analysis. From the beginning or in a later stage of the procedure? Is the analysis solely deductive? The description of data analysis needs to be further developed in order to really understand what you have done.

The data analysis explanation in the methods section has been further developed and revised to detail the point at which we mapped the themes of the study unto the levels of the SEM.

Results

Overall, the results include several important factors that are facilitators or barriers in the nurses HPB which is good and harmonize with the aim of the study. Still, the findings are really extensive and some sub-categories are overlapping each other which makes it hard to read. For example, the text on line 203-207 lies very close to the text written in the sub-category “personal experiences of life style related illness”. Another example is that time constraints are closely related and duplicated to some extent in the sub-categories “sleep deprivation, stress and fatigue” and “female gender role”. This might be due to the use of the SEM in the analysis procedure but if that is the case it needs to be clarified in the manuscript. If that not is the case I think the quality of the manuscript would improve significantly if the finding section was went through one more time and the categories and sub-categories further elaborated.

Thank you for your comment, as explained above, the findings section has been significantly revised and further developed to address the concerns you have raised.

 Discussion

Important aspects are brought up in the discussion and I think this part of the manuscript is well written.

In the paragraph focusing on methodological considerations it would be beneficial if you could elaborate on the issues I brought up in my comments regarding the method.

Thank you for your comments. We have revised the methods, findings and discussion sections accordingly.

Reviewer 4 Report

Introductory section: the acronym SEM must be correctly reported.

Materials and methods' section: the acronyms HB, RW and CU are not specified. The table is not reported in the text and results are not commented either. Besides, SEM method is not described at all. What is requested in the interview is not clear. The authors do not explain how the original number of nurses (n=210) dropped down to 18. No statistical analysis has been carried out. 

Results' section: the results should be schematized and condensed also with the use of tables and graphs.

Author Response

4th Reviewer’s comments

Authors’ responses

Introductory section: the acronym SEM must be correctly reported.

Thank you for your comments. We have now correctly reported SEM in the introduction section.

Materials and methods' section: the acronyms HB, RW and CU are not specified.

The materials and methods section have been revised to report HB, RW and CU as study authors.

The table is not reported in the text and results are not commented either.

Thank you for noticing this omission. The table has been titled as ‘table 1’ and signposted and explained in the participants’ characteristics section of the results.

Besides, SEM method is not described at all. What is requested in the interview is not clear. The authors do not explain how the original number of nurses (n=210) dropped down to 18. No statistical analysis has been carried out. 

Thank you for your comment. The SEM has been further described in later on in the introduction section.  The methods section has also been revised to include details of how and when we utilised the SEM in our study.

Also, statistical analysis would not be appropriate for our data type. The research study utilized a qualitative methodology hence statistical analysis was not performed – the nurses interviewed were from our invited sample of 210. The concept of data saturation (when further interviews yielded no new knowledge) was used to determine when to stop conducting further interviews.

Results' section: the results should be schematized and condensed also with the use of tables and graphs.

The findings of our study yielded words from transcribed interviews which would only lend themselves to qualitative data analysis. Condensing our findings into graphs and tables would not be in line with our chosen qualitative methodology.

Round 2

Reviewer 2 Report

As it looks now, it is acceptable.

Author Response

Thank you for your prompt revision. Your comments and contributions helped improve the overall quality of our manuscript.